# The Ionophores CCCP and Gramicidin but Not Nigericin Inhibit *Trypanosoma brucei* Aquaglyceroporins at Neutral pH

**DOI:** 10.3390/cells9102335

**Published:** 2020-10-21

**Authors:** Lea Madlen Petersen, Eric Beitz

**Affiliations:** Pharmaceutical and Medicinal Chemistry, Christian-Albrechts-University Kiel, 24118 Kiel, Germany; lpetersen@pharmazie.uni-kiel.de

**Keywords:** ionophore, protonophore, CCCP, nigericin, gramicidin, aquaporin, aquaglyceroporin, glycerol, pentamidine, TbAQP2, TbAQP3

## Abstract

Human African trypanosomiasis (HAT) is caused by *Trypanosoma brucei* parasites. The *T. brucei* aquaglyceroporin isoform 2, TbAQP2, has been linked to the uptake of pentamidine. Negative membrane potentials and transmembrane pH gradients were suggested to promote transport of the dicationic antitrypanosomal drug. Application of ionophores to trypanosomes further hinted at direct inhibition of TbAQP2 by carbonyl cyanide *m*-chlorophenyl hydrazone (CCCP). Here, we tested for direct effects of three classical ionophores (CCCP, nigericin, gramicidin) on the functionality of TbAQP2 and the related TbAQP3 at conditions that are independent from the membrane potential or a proton gradient. We expressed TbAQP2 and TbAQP3 in yeast, and determined permeability of uncharged glycerol at neutral pH using stopped-flow light scattering. The mobile proton carrier CCCP directly inhibited TbAQP2 glycerol permeability at an IC_50_ of 2 µM, and TbAQP3 to a much lesser extent (IC_50_ around 1 mM) likely due to different selectivity filter layouts. Nigericin, another mobile carrier, left both isoforms unaffected. The membrane-integral pore-forming gramicidin evenly inhibited TbAQP2 and TbAQP2 in the double-digit micromolar range. Our data exemplify the need for suitable controls to detect unwanted ionophore side effects even when used at concentrations that are typically recommended to disturb the transmembrane ion distribution.

## 1. Introduction

Transmembrane drug transport has been identified as a key mechanism in the generation of protozoal chemotherapy resistance [1]. Three distinct transporters from *Trypanosma brucei* have been reported to facilitate uptake of pentamidine [2]. Pentamidine is the former gold standard in the therapy of human African trypanosomiasis (HAT) [3], an infectious disease caused by *T. brucei* parasites [4]. Pentamidine (Figure 1) carries two strongly basic amidine moieties (pK_a_ 12.1) rendering the molecule permanently protonated, i.e., positively charged, under physiological conditions, which also contributes to its low oral bioavailability. In conjunction with insufficient penetration of the central nervous system, this limits the use of pentamidine to the treatment of the early, non-cerebral stages of HAT. The *T. brucei* aquaglyceroporin isoform 2, TbAQP2 (Figure 1), was shown to be the major facilitator linked to pentamidine uptake into the parasites [5]. TbAQP2 is a member of the solute channel subfamily of the ubiquitous homotetrameric aquaporins (AQP). It facilitates, besides water, the transmembrane passage of glycerol, urea, dihydroxyacetone [6], ammonia [7], arsenite and antimonite [8], lactic acid and methylglyoxal [9]. Although this substrate spectrum is common for aquaglyceroporins, the amino acid composition of the TbAQP2 selectivity filter is special [5]. Typical aquaporin selectivity filters carry an arginine residue in lipophilic, aromatic amino acids environment, referred to as the aromatic arginine region (ar/R, Figure 1). The ar/R region represents the narrowest constriction in the AQP transduction pathway [10], selects substrates by size, and electrostatically repels cations and protons [11,12]. The TbAQP2 selectivity filter, however, lacks the highly conserved arginine and, at the same time, exposes an aspartate residue to the channel lumen (Figure 1). The latter feature reverses the electrostatic surface charge from positive to negative and is unique in the AQP protein superfamily. This peculiar layout of the TbAQP2 selectivity filter in combination with the finding that pentamidine uptake into trypanosomes depends on the presence of TbAQP2 raised the question whether pentamidine might even be a direct permeant of TbAQP2 [5,13]. In fact, there are examples of drug uptake facilitated by aquaglyceroporins [14,15,16], yet such drugs resemble classical aquaglyceroporin substrates, which is not the case for pentamidine given its large size (340 Da) and positive charge. We found earlier, that pentamidine binds with nanomolar affinity to the TbAQP2 selectivity filter virtually impeding glycerol facilitation to a hold [17]. Since TbAQP2 is located in the flagellar pocket of bloodstream trypanosomes [5] that exhibits massive rates of membrane turnover cycles [18,19,20], we proposed an alternative route of uptake by endocytosis of TbAQP2-bound pentamidine [17]. 

In any case, as a cationic molecule, uptake of pentamidine into trypanosomes should benefit from the generally negative membrane potential. It has been suggested that pentamidine uptake further depends on the proton motive force established by three plasma membrane H^+^-ATPases [21], and treatment with ionophores indeed led to decreased pentamidine uptake [2,22]. Ionophores are molecules that facilitate transmembrane transport of certain ions including protons (protonophores) either by shuttling the ion across the membrane in a bound form to the ionophore, or by forming ion-permeable channels in the membrane. As a consequence, transmembrane ion or pH gradients will break down eliciting a decoupling effect. As a caveat, at least one of the protonophores used in the pentamidine experiments, carbonyl cyanide *m*-chlorophenyl hydrazone (CCCP), appeared to directly affect TbAQP2 functionality [22]. This prompted us to test for direct inhibition of TbAQP2 by common ionophores using conditions that exclude indirect effects via the disturbance of transmembrane charge distributions. We assayed TbAQP2 and the related TbAQP3 (77% identity, 87% similarity, classical ar/R region, Figure 1) in the absence of a transmembrane proton gradient, i.e., at a neutral external pH, and used glycerol as a neutral, non-protonatable substrate whose permeation is, thus, independent of proton gradients or the membrane potential. In addition to CCCP, we selected nigericin and gramicidin representing two other chemical classes of ionophores.

Here, we show direct inhibition of TbAQP2 glycerol permeability by CCCP in the single-digit micromolar range. TbAQP3 was also inhibited yet required three orders of magnitude higher concentrations. Nigericin, another mobile carrier ionophore, neither affected TbAQP2 nor TbAQ3. However, the channel-forming gramicidin directly inhibited both, TbAQP2 and TbAQP3, to a similar extent in a concentration range that is typically used in experimental designs.

## 2. Materials and Methods

### 2.1. Plasmids, Yeast Transformation and Culture

As described previously [17], we transformed yeast cells of the *Saccharomyces cerevisiae* strain BY4742Δfps1 (MATa, his3-1, leu2Δ0, lys2Δ0, ura3Δ0, yll043w::KanMX) (Euroscarf, Frankfurt, Germany) lacking the endogenous aquaglyceroporin Fps1 with pRS426Met25-tbaqp2 (GenBank: CAG27021), pRS426Met25-tbaqp3 (GenBank: CAG27022), or empty vector, using the lithium acetate method. The constructs carry an N-terminal hemagglutinin (HA) epitope tag. Transformed yeasts were grown in liquid selective media (S.D.) (0.17% yeast nitrogen base, 2% glucose, 0.5% (NH_4_)_2_SO_4_, 0.002% histidine, 0.002% lysine, 0.01% leucine) at 29 °C on an orbital shaker at 220 rpm to an OD_600_ of 1.

### 2.2. Western Blot

50 mL yeast cultures in S.D. media were harvested and disrupted using acid-washed glass beads [23]. The cell lysates were separated by SDS-PAGE and proteins were transferred to polyvinylidene fluoride membranes (Amersham Hybond-P 0.45, GE Healthcare Life Sciences, München, Germany). Chemiluminescence detection (ECL Chemostar, Intas Science Imaging Instruments, Göttingen, Germany) was carried out using a monoclonal mouse Anti-HA antibody (1:5000, Roche; cat. no. 11583816001, Mannheim, Germany), a horseradish peroxidase secondary goat anti-mouse antibody (1:5000, Jackson ImmunoResearch/Dianova; cat. no. 115-035-174, Hamburg, Germany), and the Clarity Western ECL substrate (Bio-Rad, Feldkirchen, Germany).

### 2.3. Yeast Protoplast Preparation

50 mL yeast cultures in S.D. media were harvested by centrifugation (4000× *g*, 4 °C), washed and resuspended in 2 mL buffer of 50 mM MOPS, 0.2% 2-mercaptoethanol, pH 7.2. After incubation for 15 min at 30 °C on an orbital shaker at 140 rpm, 4 mL buffer of 50 mM MOPS, 0.2% 2-mercaptoethanol, pH 7.2 supplemented with 1.8 M sucrose, as well as 120 U Zymolyase-20T (Roth, Karlsruhe, Germany) and 100 mg albumin fraction V (Roth, Karlsruhe, Germany) were added. After 1 h, protoplasts were collected at 2000× *g*, 4 °C, washed, diluted to an OD_600_ of 2 in buffer (10 mM MOPS, 50 mM NaCl, 5 mM CaCl_2_, 1.2 M sucrose, pH 7.2) and stored on ice or at 4 °C. 

### 2.4. Glycerol Permeability Assay by Stopped-Flow Light Scattering

Yeast protoplasts were rapidly mixed with an equal volume of challenging buffer in a stopped-flow apparatus (SFM-2000, BioLogic, Claix, France) with an estimated dead time of 1.6 ms, total flow rate of 14 mL s^–1^ of 150 µL, at 20 °C. For hypertonic glycerol permeability measurements, the protoplasts were challenged with buffer supplemented with 0.6 M glycerol. For isotonic conditions 0.6 M of sucrose of the buffer were replaced by an isomolar amount of glycerol. In either case, an inward gradient of 300 mM glycerol was established. Protoplast volume changes were monitored measuring the intensity of 90° light scattering at 524 nm. Signal traces (*n* = 5–9) were averaged, normalized, and single-exponentially or linearly fitted. The initial slope is a measure of solute permeability. Traces from the empty vector controls were subtracted. The glycerol permeability coefficient was calculated using the equation *P_sol_* = |*dI*/*dt*| (*V_0_*
*C_out_*)/(*S_0_*
*C_diff_*) with *d*I/dt** being the slope of the intensity curve, *V*_0_ the initial mean protoplast volume (65.45 μm^3^), *S_0_* the initial mean protoplast surface area (78.54 μm^2^), *C_out_* the total external solute concentration (1.5 M), and *C_diff_* the chemical solute gradient (0.3 M) [24].

For TbAQP2 and TbAQP3 inhibition, respective protoplasts were incubated at room temperature for 20 min prior to the assay with gramicidin (ethanolic solution; mixture of gramicidin A, B, C, and D; Sigma Aldrich, Darmstadt, Germany), nigericin (ethanolic solution; Sigma Aldrich), or carbonyl cyanide m-chlorophenyl hydrazone (CCCP, dissolved in DMSO; Sigma Aldrich). The final solvent concentrations in the samples and controls were kept at ≤1%, except for the highest gramicidin concentration where the ethanol concentration was 2%.

### 2.5. Statistical Analysis

For sample sizes ≥3, we tested the mean values of two independent data sets for significance using Student’s *t*-test (unpaired, two-tailed). If the calculated t-value was larger than the tabulated t-value (*P* = 0.95; f = n_1_ + n_2_ − 2), with n being the sample size of a population and f being the degrees of freedom, this indicated a difference between the mean values at the chosen significance level. A *t*-test is feasible even for small sample sizes [25], but requires that the two sample populations are normally distributed and have the same variance [26]. To quickly assess the normality of the samples, we used the David-Hartley-Pearson test (*P* = 0.95), which is based on the ratio of sample range to standard deviation [27] and states critical values for samples sizes as low as three [28]. Variance homogeneity was determined using the F-test (*P* = 0.95).

## 3. Results

### 3.1. Functional Expression of T. brucei Aquaglyceroprins in Yeast

We employed yeast-based aquaporin expression and permeability assays as established earlier [17]. The used yeast strain lacks the endogenous aquaglyceroporin Fps1 (BY4742Δfps1) and displays minuscule background transmembrane diffusion of glycerol. TbAQP2 and TbAQP3 were well expressed in these cells (Figure 2a). 

To assay glycerol permeability, we enzymatically removed the yeast cell wall, which renders the obtained protoplasts sensitive to osmotic stress and permits monitoring of changes in cell volume by light scattering. We assessed the general functionality of the system by challenging TbAQP2-containing protoplast suspensions by a hypertonic inward gradient of 300 mM glycerol in a stopped-flow device (Figure 2b). Under these conditions, the protoplasts will rapidly release water in an initial phase as indicated by an increase of the light scattering intensity. In the presence of aquaglyceroporins, glycerol will enter the cells in a subsequent slower phase by following the chemical inward gradient. Water that osmotically accompanies the glycerol into the protoplasts will lead to a regain of cell volume and the light scattering signal will decrease.

TbAQP2 expressing protoplasts showed the described bi-phasic light scattering curve (Figure 2b, red trace) yielding a glycerol permeability rate of *P_sol_* = 3.5 µm s^–1^. The data match earlier assays in this system [17]. Control protoplasts lacking a functional aquaglyceroporin did not measurably permit glycerol uptake as indicated by an unaltered signal intensity in the second phase of the light scattering trace (Figure 2b, black trace). With the principal functionality confirmed, for the following experiments, we switched to isotonic conditions to omit the initial osmotic protoplast shrinkage phase. The resulting light scattering curves in a 300 mM glycerol gradient were monophasic and directly indicated inward permeability (Figure 2c).

### 3.2. TbAQP2 is Directly Inhibited by CCCP, TbAQP3 Is Much Less Susceptible

Addition of the protonophore CCCP (Figure 3a) led to an effective, dose-dependent inhibition of TbAQP2 (Figure 3b, only selected concentrations are shown for clarity). Glycerol permeability of TbAQP2 was even fully blocked when CCCP was added at concentrations close to 100 µM. 

Besides TbAQP2, we tested TbAQP3, which carries a canonical aquaglyceroporin selectivity filter motif [5] (see Figure 1). TbAQP3 appeared also inhibitable by CCCP, yet required much higher concentrations that only led to partial inhibition at the highest testable CCCP concentration of 1 mM (Figure 3c, traces obtained with concentrations below 400 µM are not displayed for clarity).

To determine the IC_50_, we derived the glycerol permeability rates from the slopes of the light scattering curves in relation to that of uninhibited protoplasts and plotted the values against the concentration of CCCP. This gave a low, single-digit micromolar IC_50_ of (2.0 ± 0.3) µM of CCCP for TbAQP2 (Figure 3d). This value must be seen in connection with a typical experimental setup aiming at the breakdown of a transmembrane proton gradient at which CCCP is applied over a wide range of concentrations, usually at up to 50 µM [29,30,31], i.e., 25 times the IC_50_ of TbAQP2 inhibition. For TbAQP3, we could not completely cover the IC_50_ curve (Figure 3e). However, because the highest possible test concentration of 1 mM CCCP led to an inhibition of about 50%, the IC_50_ can be expected to lie close to this concentration.

### 3.3. Nigericin Does Not Affect Glycerol Permeability of TbAQP2 or Pentamidine Binding

We then evaluated if a second mobile carrier ionophore, nigericin (Figure 4a), is also capable of inhibiting TbAQP2 glycerol permeability. We tested nigericin (1–100 µM) on TbAQP2 expressing protoplasts, yet observed no change in the light scattering signal upon exposure to an isotonic glycerol inward gradient (Figure 4b). 

Additionally, we tested nigericin on TbAQP3 expressing protoplasts. Again, we observed no change in the light scattering signal (Figure 4c). Since nigericin, like CCCP, has been linked to inhibition of pentamidine uptake in trypanosomes [2,22], we tested if nigericin affected TbAQP2 functionality in connection to pentamidine. We showed before that pentamidine acts as a direct inhibitor of TbAQP2 glycerol permeability [17]. This inhibitory effect of pentamidine is displayed in the red trace of Figure 4d. We then preincubated TbAQP2 with nigericin and subsequently added pentamidine (isethionate salt; Sigma Aldrich) for another 10 min prior to the assay. Yet still, nigericin left the system unaffected as displayed by the blue trace in Figure 4d. In our hands, nigericin affected TbAQP2 glycerol permeability neither directly, nor via interference with pentamidine. Plotting of the relative effects of nigericin with or without pentamidine visualizes that TbAQP2 and TbAQP3 functionality remained unaltered (Figure 4e).

### 3.4. Gramicidin Inhibits TbAQP2 and TbAQP3 to a Similar Extent

Finally, we turned to gramicidin, i.e., a peptidic ionophore that, differently from CCCP and nigericin, integrates into lipid bilayers to form membrane-spanning cation-permeable channels [32,33] (Figure 5a). As a consequence, respective ion and proton gradients dissipate. In our assay, we indeed observed dose-dependent inhibition of TbAQP2. At 200 µM gramicidin, glycerol permeability was almost fully blocked (Figure 5b). 

Likewise, gramicidin inhibited TbAQP3-facilitated glycerol permeability, apparently at a similar extent as that of TbAQP2 (Figure 5c). The low water solubility of gramicidin and limitations in the maximally tolerated ethanol content in the assay prevented us from using higher concentrations to obtain full inhibition. However, plotting of the relative glycerol permeability against the gramicidin concentration provided sufficient data for sigmoidal fitting (Figure 5e). This yielded IC_50_ values of (72 ± 8) µM and (57 ± 3) µM for TbAQP2 and TbAQP3, respectively. Further statistical analysis showed that the inhibitory values for TbAQP2 and TbAQP3 at the three highest gramicidin test concentrations indicated equal inhibition (Student’s unpaired, two-tailed *t*-test; *p* > 0.05). We beforehand tested for normal distribution and variance homogeneity, which is a prerequisite for this type of analysis; see Section 2.5 for a detailed description of the statistical analysis. We conclude, that TbAQP2 and TbAQP3 are susceptible to gramicidin to a similar extent, i.e., inhibition by gramicidin is independent of the different layout in the selectivity filter region of TbAQP2 and TbAQP3.

## 4. Discussion

Ionophores are invaluable tools for studying transmembrane transport of charged substrates. At the same time, working with ionophores is a complicated matter. Not only are they chemically heterogeneous, but also their mechanisms of action, the number and direction of transported ions or protons, and their selectivity vary. In addition, the transport can be electrical or electroneutral, the latter meaning that the net transmembrane charge distribution remains equal. Therefore, when using ionophores, the experimental setup should be carefully designed. Here, we show that the use of ionophores further requires reliable controls to identify and exclude effects that are not related to the breakdown of a transmembrane ion gradient but to direct inhibition of the studied transport proteins. Of the three ionophores tested in this work, two exhibited direct inhibitory effects on the glycerol permeability of the studied TbAQPs (Figure 6). One, CCCP, acted specifically on the TbAQP2 isoform, whereas the second one, gramicidin, elicited more unspecific effects on both TbAQP2 and TbAQP3. Generally, ionophores act by two different modes [34]. First, CCCP, nigericin and related molecules function as mobile carriers. Such ionophores bind protons (covalently) or ions (noncovalently), move across the membrane, and subsequently release the transported ion on the counter side of the membrane. Second, gramicidin-like molecules form membrane-spanning channels with hydrophilic pores that facilitate transmembrane ion passage down a concentration gradient.

CCCP is used as a pH uncoupling agent in the plasma membrane, vacuoles, and mitochondria. As a weak acid (pK_a_ ≈ 6) [35] with an extended π-orbital system, both the protonated neutral and the anionic form can diffuse across the membrane. The former is driven by the proton gradient (ΔpH), the latter by the membrane potential (ΔΨ) leading to efficient cycling between the involved compartments until the gradients are eliminated [36]. At an IC_50_ of 2 µM, the potency of CCCP on the TbAQP2 glycerol permeability is remarkable, since CCCP is usually used in the two-digit micromolar concentration range. This effect is clearly independent of the action of CCCP on the transmembrane pH gradient, because neutral solute permeability of AQP channels is not affected by the proton availability [37,38]. Therefore, the CCCP inhibition of TbAQP2 must be seen as an unexpected and serious side effect that occurs at conventionally used concentrations. The different susceptibility of TbAQP2 and TbAQP3 for CCCP interaction is remarkable given their generally similar protein layout and electrostatic surface charge distribution [9]. Therefore, we relate the CCCP selectivity to the different selectivity filter layouts of TbAQP2 and TbAQP3 suggesting binding of CCCP in the peculiar ar/R region of TbAQP2 (Figure 6a).

Contrary to CCCP, nigericin left the glycerol permeability of TbAQP2 and TbAQP3 unaffected. Up to 100 µM of nigericin induced no change in the light scattering signal. Like CCCP, nigericin is a weak acid and capable of transferring protons across cellular membranes. Different from CCCP, the ionophore action of nigericin occurs without a change in the net charge. As a mobile carboxylic ionophore, nigericin is membrane permeable, both as a protonated neutral acid as well as a salt in conjunction with an inorganic cation, thus, catalyzing the electroneutral exchange of K^+^ for H^+^ and equalizing their gradients across the membrane [39]. The final ion concentrations established on both sides of the membrane depend on the experimental conditions, e.g., by pH buffering. Since our experimental setup was designed to be independent from any such effects, we can conclude that nigericin does not directly interact with TbAQP2 or TbAQP3. The much larger molecule size of nigericin compared to CCCP alone suggests that binding sites within the transduction pore of the AQPs are inaccessible for nigericin.

The even larger gramicidin peptide, on the other hand, inhibited glycerol permeability of TbAQP2 and TbAQP3 to an even degree. We expect that an interaction between gramicidin and the AQP proteins occurs outside of the pore regions and by a more unspecific mechanism (Figure 6b). Since gramicidin inserts into the membrane and extends large tryptophan aromatic sidechains (Figure 5a), it is thinkable that the AQP functionality is disturbed by intramembrane interactions with gramicidin. It has been shown before that the functionality of integral membrane proteins is affected by the lipid environment [40]. A non-specific modulating influence on TbAQP2 and TbAQP3 functionality solely by an increased loading of the membrane with gramicidin could explain the observed inhibitory effect.

Do these results affect the debate on the predominant pentamidine uptake mechanism of trypanosomes, i.e., TbAQP2-driven diffusion or endocytosis? Some aquaporins in the human system underlie a regulation mechanism based on vesicle trafficking and endocytosis to adapt the number of channels in the plasma membrane. A prominent example is the vasopressin-dependent vesicle shuttling of AQP2 in the kidney [41]; more recently, vesicle shuttling was also shown for AQP4 in astrocytes [42]. The localization of TbAQP2 in the trypanosomal flagellar pocket is of particular interest due to the exceptionally high membrane turnover rates [18,19,20]. It is clear that pentamidine acts as a potent inhibitor of the glycerol permeability of TbAQP2 but not TbAQP3 in the yeast-based system [17] as well as of TbAQP2 in trypanosomes [22]. Consistent with this, pentamidine uptake in trypanosomes was found to be mediated by TbAQP2 and not TbAQP3 [5]. A recent study put a number to the pentamidine transport rate in trypanosomes, which was estimated to be in the range of 10^6^–10^7^ molecules per cell per hour [22], i.e., extremely slow. If one assumes that the number of TbAQP2 proteins in a parasite cell is in the hundreds or low thousands, which is a cautious estimate as red blood cells carry several dozens of thousands of AQPs in the plasma membrane [43,44]. This means that the transmembrane passage of one pentamidine molecule requires about 1 s time. For comparison, the passage of a water molecule through an AQP takes place at the single-digit nanosecond time scale [45]. Solutes such as glycerol pass an aquaglyceroporin pore about 100 times slower than water [46], yet would then still require less than a microsecond to cross the membrane. This leaves a gap of six orders of magnitude between the traversing time of pentamidine and glycerol through TbAQP2. Therefore, classical AQP functionality assays e.g., light scattering will not pick up permeability rates as low as that estimated for pentamidine. In fact, such compounds would be registered as effective inhibitors of AQP water or solute permeability [17]. Generally, it will be difficult to attribute the slow transport of pentamidine exclusively to permeability or endocytosis; a joint contribution of both would appear quite realistic. Either proposed uptake pathway for the cationic pentamidine would certainly profit from the negative membrane potential. However, as this study exemplifies, caution is required when working with ionophores, and implementation of suitable controls is recommended to detect unwanted side effect.

## Figures and Tables

**Figure 1 cells-09-02335-f001:**
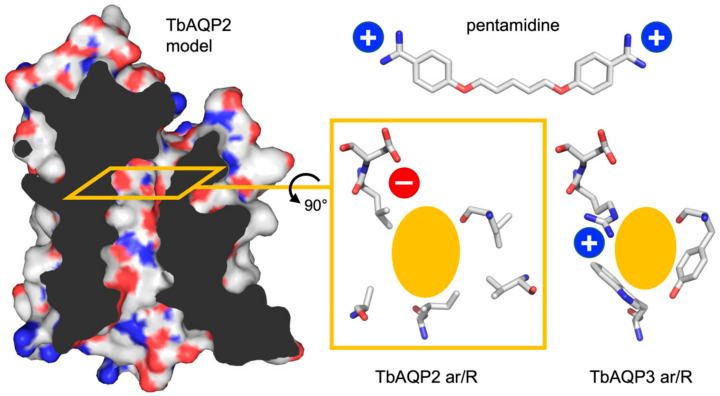
Pentamidine and structure models of TbAQP2 and TbAQP3. The TbAQP2 protomer model based on the *Escherichia coli* aquaglyceroporin GlpF (PDB #1fx8) is shown as a side view and clipped to display the water/solute transduction pathway. The plane of the selectivity filter (aromatic arginine region, ar/R) is indicated by the orange rectangle. The layouts of the TbAQP2 and TbAQP3 ar/R regions are shown in top down view and the pentamidine molecule is drawn to scale.

**Figure 2 cells-09-02335-f002:**
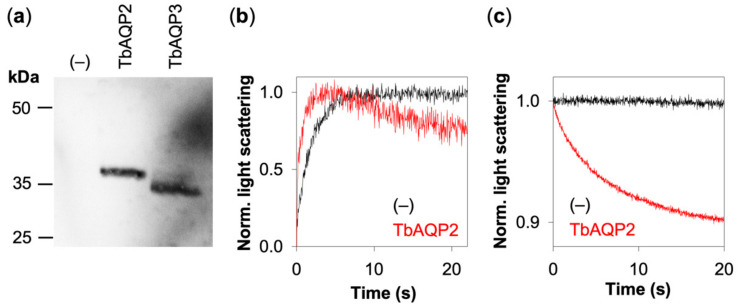
Expression of *T. brucei* aquaglyceroporins in yeast and functionality assay. (**a**) The Western blot shows TbAQP2 (35.4 kDa) and TbAQP3 (34.4 kDa) detected via an N-terminal hemagglutinin epitope tag. (**b**) Light scattering AQP functionality assay under hypertonic conditions (300 mM inward gradient of glycerol). The initial rapid phase (<5 s) indicates water efflux and protoplast volume decreases, followed by a slower phase of volume regain upon glycerol influx. (**c**) Light scattering AQP functionality assays under isotonic conditions (300 mM inward gradient of glycerol) show a monophasic curve, indicating cell swelling due to glycerol influx. The signals shown in (**b**,**c**) were averaged from 6–9 traces and normalized.

**Figure 3 cells-09-02335-f003:**
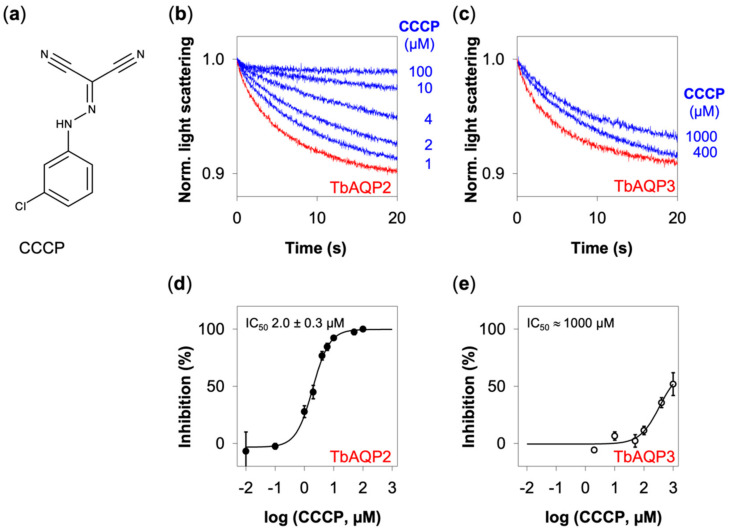
Inhibition of TbAQP2 and TbAQP3 by CCCP. (**a**) Chemical structure of CCCP. (**b**) Shown are normalized light scattering signals from TbAQP2 glycerol permeability in the absence (red trace) and presence of CCCP (blue traces). For clarity only selected CCCP concentrations are shown. (**c**) Shown are normalized light scattering signals from TbAQP3 without (red trace) and with CCCP (blue traces). For clarity concentrations below 400 µM of CCCP are not shown. The curves in (**b**,**c**) were obtained from 7–9 averaged traces and normalization. (**d**) IC_50_ curve of TbAQP2 inhibition by CCCP. (**e**) Dose-response of TbAQP3 after incubation with CCCP. Error bars indicate SEM (*n* = 2–5). In all experiments, the DMSO concentration was kept ≤1%.

**Figure 4 cells-09-02335-f004:**
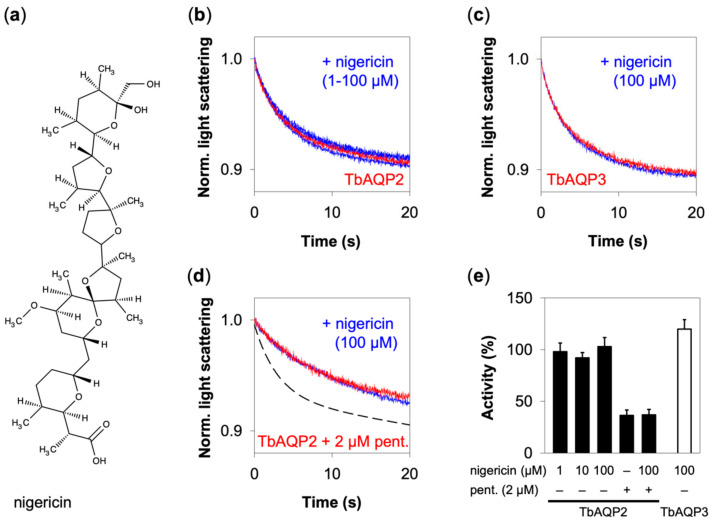
Absence of an effect of nigericin on TbAQP2 and TbAQP3. (**a**) The chemical structure of nigericin. (**b**) Shown are normalized light scattering signals from TbAQP2 glycerol permeability without (red trace) and with preincubation with nigericin (blue traces). (**c**) Normalized light scattering signals from TbAQP3 without (red traces) and with nigericin (blue traces). The curves in (**b**,**c**) were averaged from 6–9 traces and normalized. (**d**) Normalized light scattering signals from TbAQP2 that was partially inhibited by direct addition of 2 µM of pentamidine (red trace), or after pretreatment with 100 µM nigericin (blue trace). The dashed line indicates the position of the uninhibited TbAQP2 signal trace from (**b**). The curves were averaged from 5–8 traces and normalized. (**e**) Relative activity of TbAQP2 and TbAQP3 in the absence and presence of nigericin, with and without co-application of 2 µM pentamidine. Shown are mean values ± SEM (*n* = 2–3). The ethanol concentration for the solubilization of nigericin was kept at 1% in all assays.

**Figure 5 cells-09-02335-f005:**
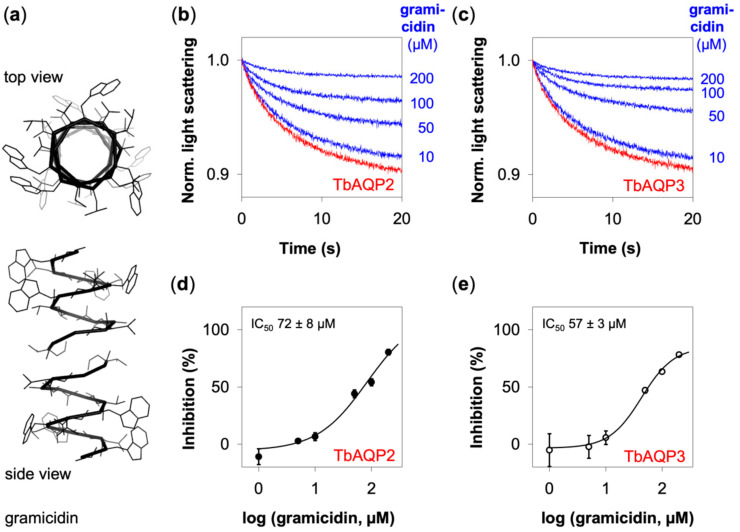
Inhibition of TbAQP2 and TbAQP3 by gramicidin. (**a**) Depictions of the gramicidin transmembrane ion channel (PDB #1mag) in a top down and side view. (**b**) Shown are normalized light scattering signals of TbAQP2 glycerol permeability without (red trace) and with gramicidin (blue traces). (**c**) Normalized light scattering curves of TbAQP3 without (red trace) and with gramicidin (blue traces). The curves in (**b**,**c**) were averaged from 8–9 traces and normalized. (**d**) IC_50_ curve of TbAQP2 inhibition by gramicidin. (**e**) IC_50_ curve of TbAQP3 inhibition by gramicidin. Shown are mean values ± SEM (*n* = 2–3). The ethanol content was kept at a final concentration of 1%, except for 200 µM gramicidin for which an ethanol concentration of 2% was necessary.

**Figure 6 cells-09-02335-f006:**
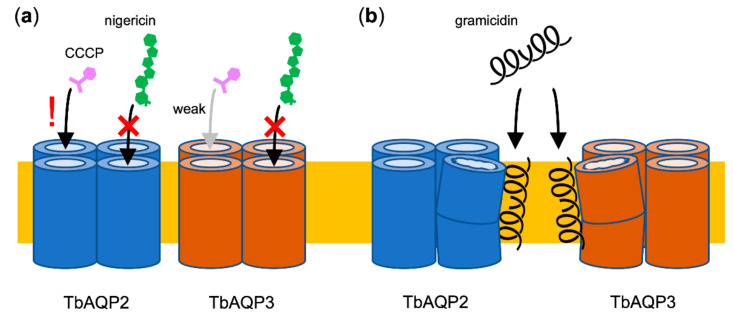
Proposed mechanisms of direct inhibition of TbAQPs by ionophores. (**a**) The small, mobile ionophore CCCP is compatible with high-affinity binding to the TbAQP2 interior but only weakly with that of TbAQP3 probably due to the differences in the selectivity filter layout. The larger nigericin does not bind to any TbAQP. (**b**) The membrane-integral, pore-forming gramicidin disturbs the fold and functionality of TbAQP2 and TbAQP3 by more unspecific means.

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
