# Peer review of "The Ionophores CCCP and Gramicidin but Not Nigericin Inhibit Trypanosoma brucei Aquaglyceroporins at Neutral pH"

_cells, 2020, doi:10.3390/cells9102335_

Round 1

Reviewer 1 Report

The authors show in their study that the ionophores CCCP and gramicidin but not nigericin inhibit aquaporins 2 and 3 of T. brucei, as they indicate in the title of the manuscript.

Technically, the study is sound and the results are clear. The study contributes to a better understanding of aquaporin biology and the mode of action of pentamidine, the gold standard for the treatment of sleeping sickness.

However, the system used for the study is an in vitro reconstituted system and studies to address and confirm the findings in a physiological system are lacking. The relevance or consequence for sleeping sickness treatment is unclear.

The study is written in a know-it-all style in some sections, which unnecessarily sounds a bit confrontative. 

Author Response

Reviewer 1

The authors show in their study that the ionophores CCCP and gramicidin but not nigericin inhibit aquaporins 2 and 3 of T. brucei, as they indicate in the title of the manuscript.

Technically, the study is sound and the results are clear. The study contributes to a better understanding of aquaporin biology and the mode of action of pentamidine, the gold standard for the treatment of sleeping sickness.

However, the system used for the study is an in vitro reconstituted system and studies to address and confirm the findings in a physiological system are lacking. The relevance or consequence for sleeping sickness treatment is unclear.

The study is written in a know-it-all style in some sections, which unnecessarily sounds a bit confrontative. 

This reviewer does not ask directly for changes of the ms. However, we have reacted to the comments by removing the first paragraph of the introduction to lessen the focus on sleeping sickness, and further added explanatory text passages in particular regarding the ionophore function to change the “know-it-all style” to be less offensive. See also our comment to reviewer 3 who expressed a similar opinion as reviewer 1.

Reviewer 2 Report

In the present paper, the authors have investigated the potential of three different ionophores (CCCP, nigericin, gramicidin) to block glycerol permeability of two different aquaporins, namely of AQP2 and AQP3 from Trypanosoma brucei parasite. To this end they have employed yeast-based aquaporin expression and permeability assays, which they have already perfected in their previos studies. The paper is well written and is a continuation of their relatively recent important discovery that pentamidine is an inhibitor of the Trypanosoma brucei AQP2 (Song et al., 2016; Plos Pathogens). In addition to verifying this finding once more, the authors draw two main conclusions; that nigericin does not affect glycerol permeability of TbAQP2 or pentamidine binding and that gramicidin inhibits TbAQP2 and TbAQP3 to a similar extent. Unfortunately, they do not dig deeper into their last finding, though they do propose a possible mechanism, which should be tested in the future.

In my opinion the paper will be of interest to researchers in the field; however, a few aspects of the paper should be improved, before it is acceptable for publication. Notably, the authors should test their findings with the appropriate statistical tests (at the end of the Results, they do state a few tests that were performed), describe these tests in the special subsection in the Materials and Methods and properly mark the significance in the Figures (e.g. in Fig. 2 they could test the time constant of the initial rapid phase and of the cell-swelling phase and in Fig. 4 e they could compare activity).

Having said that, I have two additional suggestions:

  1. If possible, it would be nice if we could see also pentamidine molecule in the yellow inset of the Figure 1, drawn to scale to demonstrate its size compared to the ar/R region.

  1. The proposed pentamidine uptake mechanism of trypanosomes, i.e. TbAQP2-driven diffusion or endocytosis is intriguing and should be tested in the future. A study that authors could refer to, although it has focused on AQP4 in astrocytes, was performed by Potokar et al. (2013; Glia).

Author Response

Reviewer 2

In the present paper, the authors have investigated the potential of three different ionophores (CCCP, nigericin, gramicidin) to block glycerol permeability of two different aquaporins, namely of AQP2 and AQP3 from Trypanosoma brucei parasite. To this end they have employed yeast-based aquaporin expression and permeability assays, which they have already perfected in their previos studies. The paper is well written and is a continuation of their relatively recent important discovery that pentamidine is an inhibitor of the Trypanosoma brucei AQP2 (Song et al., 2016; Plos Pathogens). In addition to verifying this finding once more, the authors draw two main conclusions; that nigericin does not affect glycerol permeability of TbAQP2 or pentamidine binding and that gramicidin inhibits TbAQP2 and TbAQP3 to a similar extent. Unfortunately, they do not dig deeper into their last finding, though they do propose a possible mechanism, which should be tested in the future.

In my opinion the paper will be of interest to researchers in the field; however, a few aspects of the paper should be improved, before it is acceptable for publication. Notably, the authors should test their findings with the appropriate statistical tests (at the end of the Results, they do state a few tests that were performed), describe these tests in the special subsection in the Materials and Methods and properly mark the significance in the Figures (e.g. in Fig. 2 they could test the time constant of the initial rapid phase and of the cell-swelling phase and in Fig. 4 e they could compare activity).

As suggested, we have added a new subsection and new references to the methods detailing the statistical analysis. We refrain, however, from focusing on the initial rapid water permeability phase in order not to distract from the isotonic assay conditions that were used throughout the paper. We have done such comparisons before, yet this would fit the scope of the presented ms less well. Likewise, we would like to keep the statistical analysis to the degree used to interpret the data in Figure 4 that clearly shows that there is no significant difference in the action of nigericin on TbAQP2 and TbAQP3.

Having said that, I have two additional suggestions:

  • If possible, it would be nice if we could see also pentamidine molecule in the yellow inset of the Figure 1, drawn to scale to demonstrate its size compared to the ar/R region.

We have changed the figure accordingly and drew the pentamidine molecule in the same style as the ar/R and to scale, which is a very reasonable suggestion.

The proposed pentamidine uptake mechanism of trypanosomes, i.e. TbAQP2-driven diffusion or endocytosis is intriguing and should be tested in the future. A study that authors could refer to, although it has focused on AQP4 in astrocytes, was performed by Potokar et al. (2013; Glia).

We have added a phrase on the aquaporin regulation mechanism by vesicle shuttling and endocytosis, and added references on the classical AQP2 and the more recent AQP4 findings as suggested.

Reviewer 3 Report

By using a stopped-flow light scattering method to determine the permeability of uncharged glycerol at neutral pH, the authors tested the effects of three classical ionophores (i.e., CCCP, nigericin, gramicidin) on the functionality of TbAQP2/TbAQP3, at conditions that are independent from the membrane potential or a proton gradient. These tests are displayed as main part of the article, and Figures 3-5 demonstrate the inhibition effects of CCCP, nigericin and gramicidin, respectively, on the TbAQP2 glycerol permeability. The results show that, CCCP acts specifically on the TbAQP2 isoform, gramicidin elicits more unspecific effects on both TbAQP2 and TbAQP3, while nigericin leaves both isoforms unaffected.

Although the experimental design seems systematic, there are some shortcomings that significantly reduce the convincingness of the article. For example,

  1. The background is too much overstated. The introduction section tends to focus on the HAT therapy, pentamidine uptake, TbAQP2 selectivity filter, and at last ionophores. However, throughout the results, only the inhibition effects of three ionophores on the glycerol transport are tested.
  2. In the discussion section, the authors mentioned that “ionophores act by two different modes…CCCP, nigericin …function as mobile carriers. … gramicidin-like molecules form membrane-spanning channels with hydrophilic pores that facilitate transmembrane ion passage down a concentration gradient.” It seems that, the experiments and data are not strong enough to support the conclusion.

Therefore, the manuscript is not suggested to be published at present version. It needs to be rewritten in a more proper mode or complemented with more experiments.

Author Response

Reviewer 3

By using a stopped-flow light scattering method to determine the permeability of uncharged glycerol at neutral pH, the authors tested the effects of three classical ionophores (i.e., CCCP, nigericin, gramicidin) on the functionality of TbAQP2/TbAQP3, at conditions that are independent from the membrane potential or a proton gradient. These tests are displayed as main part of the article, and Figures 3-5 demonstrate the inhibition effects of CCCP, nigericin and gramicidin, respectively, on the TbAQP2 glycerol permeability. The results show that, CCCP acts specifically on the TbAQP2 isoform, gramicidin elicits more unspecific effects on both TbAQP2 and TbAQP3, while nigericin leaves both isoforms unaffected.

Although the experimental design seems systematic, there are some shortcomings that significantly reduce the convincingness of the article. For example,

  • The background is too much overstated. The introduction section tends to focus on the HAT therapy, pentamidine uptake, TbAQP2 selectivity filter, and at last ionophores. However, throughout the results, only the inhibition effects of three ionophores on the glycerol transport are tested.

Reviewer 1 expressed the same feeling and we agree. We have removed the first paragraph of the introduction to lessen the focus on sleeping sickness. Further, we extended explanations on the modes of action of the ionophores/protonophores.

  • In the discussion section, the authors mentioned that “ionophores act by two different modes…CCCP, nigericin …function as mobile carriers. … gramicidin-like molecules form membrane-spanning channels with hydrophilic pores that facilitate transmembrane ion passage down a concentration gradient.” It seems that, the experiments and data are not strong enough to support the conclusion.

See above, we have added text and three more reference to to better explain and illustrate the well-established function and mode of action of the different ionophore types. The references further give examples of the used CCCP concentrations in a typical experimental setup, which ranges up to 50 µM, sometimes even 100 µM, and thus much higher than our obtained IC50 for TbAQP2 inhibition of 2 µM. This notion, and the effect of gramicidin on the general functionality of TbAQP2 and TbAQP3 was the motivation for the paper pointing out a big caveat at the use of ionophores.

Therefore, the manuscript is not suggested to be published at present version. It needs to be rewritten in a more proper mode or complemented with more experiments.

We sincerely hope that our clarifications now satisfy the reviewer.

Round 2

Reviewer 2 Report

In the resubmitted (revised) manuscript, Petersen and Beitz have adequately addressed my concerns and improved the manuscript. Overall, I am sure that this paper will be of interest to researchers in the field and, in my opinion, should be accepted in the present form.

I would like to congratulate the authors for their work and wish them all the best in these challenging times.

Reviewer 3 Report

The authors have tried to answer the referee's comments. The manuscript would be helpful for the researchers in this field and can now be accepted for publication.